# Magnitude of Dural Tube Compression Still Does Not Show a Predictive Value for Symptomatic Lumbar Spinal Stenosis for Six-Year Follow-Up: A Longitudinal Observation Study in the Community

**DOI:** 10.3390/jcm11133668

**Published:** 2022-06-25

**Authors:** Koji Otani, Shin-ichi Kikuchi, Takuya Nikaido, Shin-ichi Konno

**Affiliations:** Department of Orthopaedic Surgery, Fukushima Medical University School of Medicine, Fukushima 960-1295, Japan; sinichk@db3.so-net.ne.jp (S.-i.K.); tnikaido@fmu.ac.jp (T.N.); skonno@fmu.ac.jp (S.-i.K.)

**Keywords:** lumbar spinal stenosis, epidemiology, natural history, dural tube compression, prognostic factors

## Abstract

Background: Lumbar spinal stenosis (LSS) is a clinical syndrome based on anatomic narrowing of the spinal canal. It is well known that anatomic narrowing of the spinal canal is essential for manifestation, but not all of them cause symptoms. There are many studies assessing the relationship between dural tube compression on MRI and clinical symptoms; however, most of them are cross-sectional. The purpose of this study was to reveal the magnitude of dural tube compression’s influence on the presence or development of LSS symptoms at the six-year follow-up and the occurrence of surgery during the follow-up period or not in the community setting. Methods: This was a longitudinal observational study of 459 participants who were assessed for typical LSS symptoms, and whose Roland–Morris Disability Questionnaire and numerical rating scale of leg pain and numbness was recorded using a questionnaire and conventional MRI of the lumbar spine. Typical LSS symptoms were judged using an LSS diagnostic support tool, which was a self-administered, self-reported history questionnaire (LSS-SSHQ). After six years, 232 subjects (follow-up rate 50.5%) were followed-up with typical LSS symptoms using LSS-SSHQ by mail. The relationship between the magnitude of dural tube compression evaluated by dural tube cross-sectional area (DCSA) in the initial assessment and the time course of typical LSS symptoms for the six-year duration were analyzed. In addition, predictors of the presence of typical LSS symptoms at the six-year follow-up were assessed. Furthermore, we investigated the relationship between typical LSS symptoms and DCSA during the initial assessment of patients who underwent surgery during the follow-up period. A multivariate logistic regression analysis was performed for statistical analysis. Results: (1) Severe dural tube compression did not show that LSS symptoms continued after six years. (2) Severe dural tube compression could not detect development of LSS-symptoms and surgery during the six-year period. Conclusion: Severe dural tube compression could not detect typical LSS symptom development and occurrence of surgery during the six-year period.

## 1. Introduction

Lumbar spinal stenosis (LSS) is a clinical syndrome based on anatomic narrowing of the spinal canal [1]. It is well known that anatomic narrowing of the spinal canal is essential for manifestation, but not all of them cause symptoms. CT and MRI studies of volunteers who initially had no low back pain or leg pain often showed disc degeneration (bulging, protrusion), facet joint arthropathy, or spinal stenosis with age [2,3,4]. There has been a long-standing debate about whether the degree of dural tube compression affects LSS symptom severity and quality of life; however, a conclusion has not been reached [5,6,7,8,9,10,11,12,13,14,15,16,17,18,19,20,21]. Given that the presence of dural compression is required but not solely responsible for the development of LSS symptoms, the degree of dural compression does not necessarily have to correlate with the presence and/or LSS symptom severity or Qol, at least in a cross-sectional study. 

In contrast to cross-sectional studies, few long-term longitudinal studies have examined the relationship between LSS symptoms and the degree of dural tube compression. In a follow-up study by Minamide et al., the course of clinical symptoms of 34 LSS patients over 10 years was divided into three types, namely deterioration (38%), no change (31%), and improvement (31%). In this study, six of the nine patients who underwent surgery were reported to have a compressed dural tube area of <50 mm^2^ [22]. However, the question of how dural tube compression causes symptoms still remains unresolved.

Previously, we reported cross-sectional and one-year follow-up of dural tube compression on magnetic resonance image (MRI) and clinical LSS symptoms. This report revealed that, (1) severe dural tube compression had a strong influence on the presence of typical LSS symptoms; however, 40–70% of participants with severe dural tube compression did not show typical clinical LSS symptoms, (2) magnitude of the dural tube compression did not directly affect the presence of typical LSS symptoms at the one-year follow-up [23]. Because LSS is a syndrome based on age-related degenerative changes, a one-year follow-up period may be too short. Therefore, we planned a long-term follow-up. The purpose of this study was to clarify if magnitude of dural tube compression influences the presence or development of typical LSS symptoms of six-year follow-up and the occurrence of surgery during the follow-up period in the community setting in which one-year follow-up was previously reported.

## 2. Materials and Methods

### 2.1. Initial Assessment

Details can be found in the previously published paper [5]. In brief, 459 subjects (male: 148, female: 311) who were assessed for typical LSS symptoms using a questionnaire and that received conventional MRI of the lumbar spine in 2004 were recruited (Figure 1). Subjects were recruited from the respondents to an announcement of assessment of lumbar spinal stenosis (LSS) that was part of a public health survey being conducted by their local governments. All subjects were self-sufficient (living in their own houses without the need for supplemental care and walking independently with or without support such as a cane or a walker) [23,24,25,26]. Subjects were excluded if they were unable to fill out the questionnaires by themselves, had ever undergone brain or spinal surgery, or had experienced a fracture of the lower extremities in the year prior to the start of the study period.

Typical LSS symptoms were judged using a specially designed and validated LSS diagnostic support tool, which is a self-administered, self-reported history questionnaire (LSS-SSHQ) [27]. For evaluation of leg-symptom intensity, an 11-point numerical rating scale (NRS; 0: absence of pain/numbness, 10: worst pain/numbness) was used. For assessment of low back pain (LBP)-related quality of life (QoL), the Roland–Morris Disability Questionnaire (RDQ) (Japanese version) was assessed [28,29,30,31]. A Japanese version of the RDQ established the national normative RDQ score and standard deviation values (available for 20–79 years old, male/female). Fifty points are the national average, while more than 50 points indicate a higher-than-average LBP-related QoL and less than 50 points indicate a lower-than-average LBP-related QoL [31]. LSS-SSHQ, NRS of leg symptoms, and RDQ were collected by self-administrated questionnaire.

Axial T2-weighted images were obtained at the midpoint of each intervertebral disc from L1-2 to L5-S1 using three MRI machines. The dural tube cross-sectional area (DCSA) of L1-2–L5-S1 on the T2-weighted image was measured using the conventional formula described by Hamanishi [32]. Details of the MRI machine, imaging protocol, and DCSA measurement were described in a previous paper [23]. The smallest DCSA of L1-2–L5-S1 in each participant was divided into five categories: less than 25 mm^2^, 25–49.9 mm^2^, 50–74.9 mm^2^, 75–99.9 mm^2^, and 100 mm^2^ or more. The number of intervertebral discs of which the smallest DCSA was less than 50 mm^2^ (ranged 0–5) was also analyzed.

All participants provided written informed consent.

### 2.2. Six-Year Follow-Up

Six years later after the initial survey, six-year follow-up survey by mail was carried out. In this follow-up survey, 232 (male: 73, female: 159) of 459 subjects who participated in the initial survey responded and were assessed for typical LSS symptoms using LSS-SSHQ and six years of lumbar surgery history (follow-up rate 50.5%) (Figure 1, Table 1). Presence or absence of typical LSS symptoms was judged as the same manner of the initial survey. In addition, in this follow-up, eight subjects who received surgery for LSS during the six-year period were judged as typical LSS-positive in the six-year follow-up regardless of the judgment of LSS-SSHQ. Subjects were excluded if they were unable to walk independently and to fill out the questionnaires by themselves at the follow-up time. Subjects were also excluded if they had ever undergone brain or cervical/thoracic surgery or had experienced a fracture of the lower extremities during periods from the initial survey to the six-year follow-up.

### 2.3. Statistics

A multivariate logistic regression analysis was performed with the presence of typical LSS symptoms at the six-year follow-up as the dependent variable, and age, sex, norm-based RDQ score, NRS of leg symptoms, presence of typical LSS symptoms at the initial survey, the smallest DCSA, and the number of intervertebral discs of which DCSA was less than 50 mm^2^ as independent variables. All statistical analyses were performed using the STAT View software package (version 5.0, SAS Institute Inc., Cary, NC, USA). A p value of less than 0.05 was considered statistically significant. 

### 2.4. Ethical Approval

This study was approved by the ethical committee of Fukushima Medical University (No. 295, 673).

## 3. Results

### 3.1. LSS Symptoms Evaluated by Questionnaire at the Six-Year Follow-Up

Two-hundred and thirty-two follow-up subjects were divided into two groups, a typical LSS symptoms group and no typical LSS symptoms group at the initial survey, and changes in typical LSS symptoms at follow-up were assessed (Table 2). In the initial typical LSS-positive group, 13 of 52 subjects (25%) remained typical LSS-positive at the six-year follow-up, whereas 39 of 52 (75%) were reclassified as typical LSS-negative. On the other hand, 135 of 180 subjects (73.9%) in the typical LSS-negative group in the initial survey were still in the typical LSS-negative group at the six-year follow-up, with the other 47 of 180 (26.1%) having been reclassified as typical LSS-positive. This pattern was almost the same as the one-year follow-up [23]. The LSS-negative group was divided into subgroups: one was the no LSS-symptoms group with none of the 10 items of the questionnaire, and another was insufficient LSS-symptoms group which answered ‘yes’ for some questions but did not meet the LSS-positive criteria in LSS-SSHQ; those changes over time are shown in Appendix A.

### 3.2. Changes of Typical LSS Symptoms from the Initial Analysis to Six-Year Follow-Up

Severe dural tube compressions such as less than 25 mm^2^ and 25–49.9 mm^2^ of DCSA did not show that typical LSS symptoms continued after six years (less than 25 mm^2^; 2/16 = 12.5%, 25–49.9 mm^2^; 3/12 = 25%). Rather, typical LSS symptoms often seemed to persist with wider DCSA (75–99.9 mm^2^; 4/9 = 44.4%, 100 or more than 100 mm^2^; 3/8 = 37.5%) (Table 3). On the other hand, those with severe dural tube compression such as less than 25 mm^2^ of DCSA seemed to develop typical LSS symptoms (3/10 = 30%); however, we could not detect a statistically significant relationship between LSS-symptom development and the smallest DCSA (25–49.9 mm^2^; 8/42 = 19.0%, 50–74.9 mm^2^; 14/42 =33.3%, 75–99.9 mm^2^; 8/39 = 20.5%, 100 or more than 100 mm^2^; 14/47 = 29.8%) (Table 3). Changes in the subgroups of the typical LSS-negative group are shown in Appendix A.

The number of intervertebral discs whose DCSA was less than 50 mm^2^ did not show any tendency of maintaining of typical LSS symptoms (0; 8/24 = 33.3%, 1; 2/13 = 15.4%, 2; 3/10 = 30%, 3 or more than 3; 0/5 = 0%) (Table 4). Similarly, the number of intervertebral discs whose DCSA was less than 50 mm^2^ did not show any tendency of development of LSS symptoms (0; 36/127 = 28.3%, 1; 8/36 = 22.2%, 2; 2/12 = 16.7%, 3 or more than 3; 1/5 = 25%) (Table 4). The subgroups of the typical LSS-negative group are shown in Appendix A.

### 3.3. Analysis of Predictors for Typical LSS Symptom-Positive at the Six-Year Follow-Up

According to a multivariate logistic regression analysis, there were no predictive factors for the presence of LSS symptom by Models 1, 2, 3, and 4 (Table 5). Model 4 was the same analysis as the one-year follow-up study. Models 1, 2, and 3 were the combination of independent variables. On the other hand, at the one-year follow-up, the presence of LSS symptoms (odds ratio 4.480) and the score below the normative RDQ score (odds ratio 5.169) had a statistically significant influence on the presence of LSS symptoms not but DCSA [23]. In both one- and six-year follow-up, DCSA was not a predictive factor for the presence of LSS symptoms.

### 3.4. Characteristics of Surgery Cases during Six-Year Period

Eight subjects received surgery for LSS during the six-year period (Table 6). Magnitude of dural tube compression seemed not to influence the occurrence of surgery. 

## 4. Discussion

Our previous study pointed out some controversial issues regarding LSS etiology such as that severe dural tube compression in lumbar spine did not always show LSS symptoms, LSS symptoms seemed not to be stable for a one-year period, and the magnitude of dural tube compression was not a predictive factor for the presence of LSS at the one-year follow-up [5]. From these results, another question has arisen, that is, whether a longer follow-up period could change this result or not. In order to resolve this question, six-year follow-up surveys were carried out.

In the present study, severe dural tube compression such as those less than 25 mm^2^ and 25–49.9 mm^2^ of DCSA did not influence either persistent or developed typical LSS symptoms for six years. For example, in the <25 mm^2^ of DCSA group, persistent typical LSS symptoms were present in 12.5% (2/16) and newly developed typical LSS symptoms occurred in 30% of participants (3/10) (Table 3). Similarly, the number of intervertebral discs whose DCSA was less than 50 mm^2^ did not show any tendency of maintaining and developing typical LSS symptoms. For example, in more than three intervertebral disc levels less than 50 mm^2^ of the DCSA group, persistent typical LSS symptoms were present in 0% (0/5) and newly developed typical LSS symptoms occurred in 20% (1/5) (Table 4). Moreover, according to a multivariate logistic regression analysis of four models, magnitude of dural tube compression was not a predictive factor for the presence of typical LSS symptoms at the six-year follow-up (Table 5). This was consistent with the result of the one-year follow-up [5]. In contrast, the presence of typical LSS symptoms and low RDQ scores at the initial survey were predictors of the presence of typical LSS symptoms at the one-year follow-up, but not predictors at the six-year follow-up. This means that the degree of dural tube compression is strong enough to cancel out the positive effects of both the presence of typical LSS symptoms and low norm-based RDQ score at the initial assessment predicting the presence of LSS symptoms after six years. Therefore, dural tube compression is not a predictor of the presence of typical LSS symptoms after six years. 

It is well known that anatomical dural tube compression is necessary but not sufficient for finally diagnosing LSS symptoms because dural tube compression does not always induce symptoms [1,2,3,4]. Similarly, it is still controversial whether magnitude of dural tube compression and the severity of symptoms or Qol are correlated or not [5,6,7,8,9,10,11,12,13,14,15,16,17,18,19,20,21]. One of the reasons why anatomical dural tube compression does not induce LSS symptoms is that it might need other factors related to the development of LSS symptoms, except for anatomical dural tube compression. In degenerative lumbar spine, up to now, there is no solution to protect and/or dramatically reduce degenerative change. So, if these other factors regarding development of LSS symptoms except for anatomical dural tube compression might be controlled, LSS symptoms could be controlled. Another reason is that dural tube compression cannot detect lateral recess stenosis [14,33]. 

This study has several limitations [23]. First, the follow-up rate of 50.5% was low. Second, MRI findings did not assess lateral stenosis and foraminal stenosis. Third, LSS-SSHQ could not evaluate the severity of LSS symptoms, but just define the presence or absence of typical LSS symptoms. In addition, there were no objective data such as those obtained by electromyography. Fourth, there was no evaluation of comorbidities such as internal medicine problem, hip and/or knee osteoarthritis and more. Finally, all subjects were volunteers in a rural and mountainous area. These mean that there was a possibility of selection bias compared with the general population. In spite of these limitations, the present study might still have worth because according to our knowledge, the current study is the largest to assess the relationship between anatomical dural tube compression and typical LSS symptoms. This study shows that the magnitude of dural tube compression was not equal to the occurrence of typical LSS symptoms even if at the six-year follow-up in the community. Further study is still needed to investigate the occurrence of typical LSS symptoms in a long-term follow-up and risk factors of typical LSS symptoms for keeping health in the elderly.

## 5. Conclusions

The magnitude of dural tube compression did not influence the presence of typical LSS symptoms at the six-year follow-up and occurrence of surgery during the six-year period. Dural tube compression is an anatomical cause for LSS; however, LSS is not always symptomatic even for the six-year observation duration. Severe dural compression did not always induce surgery directly. The relationships between symptomatic and asymptomatic LSS in people with dural tube compression still remain unclear.

## Figures and Tables

**Figure 1 jcm-11-03668-f001:**
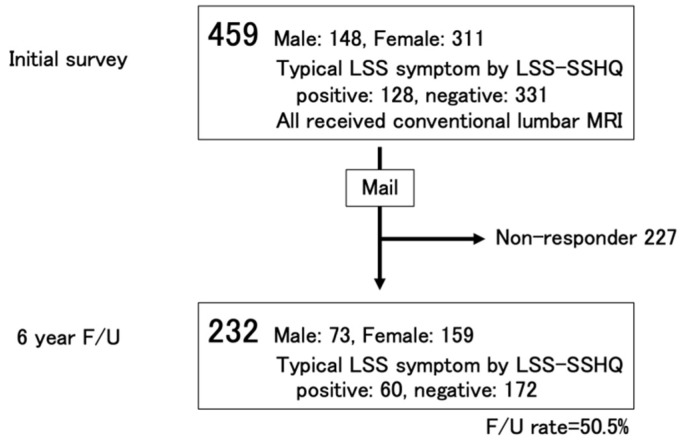
Participants.

**Table 1 jcm-11-03668-t001:** Proportion of participants at the initial survey and six-year follow-up.

Age (Years) at the Initial Survey	Initial Survey		Six-Year Follow-Up	
	Participants (n)	Rate (%)	Participants (n)	Rate (%)
<40	18	1.2	9	0.6
40–49	24	2.1	12	1.1
50–59	74	5.5	33	2.5
60–69	151	9.8	82	5.3
70–79	160	9.3	84	4.9
80–	32	3.0	12	1.1
Total	459	5.6	232	2.8

Note: There was no statistical difference of age proportion between the initial survey and six-year follow-up.

**Table 2 jcm-11-03668-t002:** Time course of LSS symptoms.

		Six-Year Follow-Up		Total
		Typical LSS-Positive	Typical LSS-Negative	
Initial analysis	Typical LSS-positive	13	39	52
	Typical LSS-negative	47	133	180
Total		60	172	232

Note: 13 of 52 subjects (25%) in the initial typical LSS-positive group remained typical LSS-positive and 47 of 180 (26.1%) in the initial typical LSS-negative group were reclassified as typical LSS-positive at the six-year follow-up. Abbreviations: LSS, Lumbar Spinal Stenosis.

**Table 3 jcm-11-03668-t003:** Relationship between change of typical LSS symptoms and the smallest DCSA.

		The Smallest DCSA (mm^2^)				
		<25	25–49.9	50–74.9	75–99.9	100≤
Changes of typical LSS symptoms from the initial survey to six-year follow-up	Positive → Positive	2	3	1	4	3
	Positive → Negative	14	9	6	5	5
	Negative → Positive	3	8	14	8	14
	Negative → Negative	7	34	28	31	33
Total		26	54	49	48	55

Note: In the relationship between the smallest DCSA and changes of typical LSS symptoms from initial survey to six-year follow-up, severity of dural tube compression did not directly influence development, improvement, and continuing of typical LSS symptoms. Abbreviations: DCSA, Dural sac Cross-Sectional Area, Av., Average, CI, Confidence Interval, LSS, Lumbar Spinal Stenosis.

**Table 4 jcm-11-03668-t004:** Relationship between change of typical LSS symptoms and the number of intervertebral discs whose DCSA was less than 50 mm^2^.

		The Number of Intervertebral Discs Whose DCSA Was Less Than 50 mm^2^			
		0	1	2	3≤
Changes of typical LSS symptoms from the initial survey to six-year follow-up	Positive → Positive	8	2	3	0
	Positive → Negative	16	11	7	5
	Negative → Positive	36	8	2	1
	Negative → Negative	91	28	10	4
Total		151	49	22	10

Note: In the relationship between the number of intervertebral discs whose DCSA was less than 50 mm^2^ and changes of typical LSS symptoms from initial survey to six-year follow-up, severity of dural tube compression did not directly influence development, improvement, and continuing of typical LSS symptoms. Abbreviations: DCSA, Dural sac Cross-Sectional Area, LSS, Lumbar Spinal Stenosis.

**Table 5 jcm-11-03668-t005:** A multivariate logistic regression analysis of predictive factors for the presence of LSS symptoms at the six-year follow-up.

		Model 1		Model 2		Model 3		Model 4	
		OR (95% CI)	*p*	OR (95%CI)	*p*	OR (95% CI)	*p*	OR (95% CI)	*p*
Age (years)		1.002 (0.975–1.029)	0.9059	0.990 (0.930–1.053)	0.6458	0.987 (0.930–1.047)	0.6535	0.991 (0.931–1.054)	0.7664
Gender	Male	ref.	-	ref.	-	ref.	-	ref.	-
	Female	1.383 (0.712–2.689)	0.3385	1.964 (0.446–8.640)	0.4047	1.652 (0.429–6.361)		2.082 (0.470–9.219)	0.3339
Norm-based RDQ score (50 = normative value)	≥50			ref.	-	ref.	-	ref.	-
	<50			1.440 (0.373–5.555)	0.5967	1.578 (0.428–5.815)	0.4931	1.382 (0.356–5.366)	0.6398
NRS of leg pain/numbness				0.852 (0.668–1.087)	0.1976	0.833 (0.659–1.054)	0.1282	0.829 (0.640–1.073)	0.1543
Typical LSS symptoms at the initial analysis	Negative			ref.	-	ref.	-	ref.	-
	Positive			0.918 (0.209–4.032)	0.9098	1.006 (0.242–4.176	0.9934	1.034 (0.228–4.699)	0.9654
The smallest DCSA		1.000 (0.989–1.010)	0.9377			1.008 (0.994–1.023)	0.2719	1.011 (0.986–1.038)	0.3929
No. of DCSA of less than 50 mm^2^	0	ref.	-	ref.	-			ref.	-
	1	0.585 (0.216–1.582)	0.2903	0.493 (0.104–2.346)	0.3741			0.995 (0.103–9.267)	0.9967
	2	0.710 (0.199–2.528)	0.5969	1.767 (0.286–10.928)	0.5403			4.041 (0.278–58.785)	0.3066
	≥3	0.264 (0.028–2.516)	0.2469	4.127 × 10^−8^ (0.000)	0.9966			1.048 × 10^−7^ (0.000)	0.9968
R^2^		0.016		0.115		0.078		0.125	

Note: There were no predictive factors for the presence of typical LSS symptoms at the six-year follow-up. Abbreviations: OR, Odds Ratio, CI, Confidence Interval, RDQ, Roland–Morris Disability Questionnaire, NRS, Numerical Rating Scale, LSS, Lumbar Spinal Stenosis, DCSA, Dural sac Cross-Sectional Area.

**Table 6 jcm-11-03668-t006:** Eight cases of surgery during the six-year period.

Initial Survey					
Case No.	Age	Gender	LSS Symptoms	The Smallest DCSA (mm^2^)	No. of DCSA of Less Than 50 mm^2^
18	83	M	Negative	118.7	0
70	75	F	Positive	107.5	0
79	72	F	Negative	46.5	1
301	74	F	Positive	58.8	0
304	67	F	Positive	94.8	0
338	65	F	Negative	51.8	0
355	65	F	Negative	55.7	0
413	59	M	Negative	63.0	0

Note: We could not detect the possible factors for surgery during six-year period. Abbreviations: LSS, Lumbar Spinal Stenosis, DCSA, Dural sac Cross-Sectional Area.

## Data Availability

The data presented in this study could be available on request from the corresponding author. The data are not publicly available because the underlying data was obtained from the collaboration with the local government and contains sensitive information on individuals including gender, age and self-reported data, and sharing these data openly is prohibited by the local government and Fukushima Medical University Ethics Committee.

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
