# Peer review of "Magnitude of Dural Tube Compression Still Does Not Show a Predictive Value for Symptomatic Lumbar Spinal Stenosis for Six-Year Follow-Up: A Longitudinal Observation Study in the Community"

_jcm, 2022, doi:10.3390/jcm11133668_

Round 1
Reviewer 1 Report
In this prospective cohort study, the authors evaluated whether magnitude of dural tube compression influences the development or presence of lumbar spinal stenosis (LSS) symptoms at the six-year follow-up. This study is a continuation of authors previously published study that had a one-year follow-up period. One of my concerns is that the authors need to be clear and consistent in reporting all the outcome measures in the abstract, objectives, methods, discussion, and conclusion. For e.g., in the abstract, the authors state that the purpose is to investigate compression influence on LSS symptoms, while conclude about its influence on LSS symptom and occurrence of surgery. Other specific comments are as follows:
Abstract:
- Include all the outcome measures mentioned in the manuscript e.g., LSS-SSHQ, NRS, RDQ for quality of life and occurrence of surgery during six-year period.
- Provide brief information on the analysis method used.
Introduction:
- The introduction section needs more work. Please provide the rational and significance of conducting the study along with what is already known and not known in the literature on this topic.
- Page 1 Ln 31, 32: Unclear sentence. Please provide more information. Reference 1 is missing from references. Also, recheck the reference numbers throughout the manuscript as the numbers do not match with the articles mentioned in the references.
Methods:
- Page 2: Please provide more detail about the inclusion criteria for the study. How did the authors screen participants for lumbar spinal stenosis? Was it based on the questionnaire or MRI or both?
Results:
Page 3, Ln 101-103: Information on participants being grouped as LSS-positive and LSS-negative appears in the results for the first time. Please define these terms in the methods section first.
Discussion:
Page 6, Ln 181-187: The reasons provided are unclear. Please provide more detail.
Author Response
Thank you very much for kind and useful comments to improve our manuscript. We considered the reviewers' intentions and made corrections to improve the paper.
Reviewer 1.
In this prospective cohort study, the authors evaluated whether magnitude of dural tube compression influences the development or presence of lumbar spinal stenosis (LSS) symptoms at the six-year follow-up. This study is a continuation of authors previously published study that had a one-year follow-up period. One of my concerns is that the authors need to be clear and consistent in reporting all the outcome measures in the abstract, objectives, methods, discussion, and conclusion. For e.g., in the abstract, the authors state that the purpose is to investigate compression influence on LSS symptoms, while conclude about its influence on LSS symptom and occurrence of surgery.
Other specific comments are as follows:
Abstract:
- Include all the outcome measures mentioned in the manuscript e.g., LSS-SSHQ, NRS, RDQ for quality of life and occurrence of surgery during six-year period.
- Provide brief information on the analysis method used.
Reply: Thank you very much for your comments. We carefully checked the consistency between the abstract and the text, and made major corrections according to the reviewer's suggestions.
Methods in Abstract
・This was a longitudinal observational study of 459 participants who were assessed for typical LSS symptoms, and whose Roland-Morris Disability Questionnaire (RDQ) and numerical rating scale of leg pain and numbness was recorded using a questionnaire and conventional MRI of the lumbar spine. Typical LSS symptoms were judged using an LSS diagnostic support tool, which was a self-administered, self-reported history questionnaire (LSS-SSHQ).
・The relationship between magnitude of dural tube compression evaluated by dural tube cross-sectional area (DCSA) in the initial assessment and the time course of typical LSS symptoms for six-year duration were analyzed. In addition, predictors of the presence of typical LSS symptoms at the six-year follow-up were assessed. Furthermore, we investigated the relationship between typical LSS symptoms and DCSA during the initial assessment of patients who underwent surgery during the follow-up period. A multivariate logistic regression analysis was performed for statistical analysis.
Introduction:
- The introduction section needs more work. Please provide the rational and significance of conducting the study along with what is already known and not known in the literature on this topic.
Reply: Thank you very much for comments. We focused the relationship between the magnitude of dural tube compression and symptoms cross-sectionally and longitudinally.
We added the sentences in Introduction;
There has been a long-standing debate about whether the degree of dural tube compression affects LSS symptom severity and quality of life; however, a conclusion has not been reached [5–21]. Given that the presence of dural compression is required but not solely responsible for the development of LSS symptoms, the degree of dural compression does not necessarily have to correlate with the presence and / or LSS symptom severity or Qol at least in a cross-sectional study.
In contrast to cross-sectional studies, few long-term longitudinal studies have examined the relationship between LSS symptoms and the degree of dural tube compression. In a follow-up study by Minamide et al., the course of clinical symptoms of 34 LSS patients over 10 years was divided into three types; deterioration (38%), no change (31%), improvement (31%). In this study, six of the nine patients who underwent surgery were reported to have a compressed dural tube area of < 50 mm2 [22]. However, the question of how dural tube compression cause symptoms still remain unresolved.
- Page 1 Ln 31, 32: Unclear sentence. Please provide more information. Reference 1 is missing from references. Also, recheck the reference numbers throughout the manuscript as the numbers do not match with the articles mentioned in the references.
Reply: Thank you very much for comments. Firstly, we apologize for the incorrect article number quoted. This is because the first paper cited in the automatic number choreography was 2 instead of 1. This is a completely careless mistake.
We changed ref. No. and one sentence; Lumbar spinal stenosis (LSS) is a clinical syndrome based on anatomic narrowing of the spinal canal [1]. It is well known that anatomic narrowing of the spinal canal is essential for manifestation, but not all of them cause symptoms. CT and MRI studies of volunteers who initially had no low back pain or leg pain often showed disc degeneration (bulging, protrusion), facet joint arthropathy, or spinal stenosis with age [12–4].
Methods:
- Page 2: Please provide more detail about the inclusion criteria for the study. How did the authors screen participants for lumbar spinal stenosis? Was it based on the questionnaire or MRI or both?
Reply: Thank you very much for comments. Original 459 participants were volunteer with/without LSS symptoms. So, there was no screening at the initial survey. Figure inserted to make it easier for the reader to understand.
Results:
Page 3, Ln 101-103: Information on participants being grouped as LSS-positive and LSS-negative appears in the results for the first time. Please define these terms in the methods section first.
Reply: Thank you very much for comments. In the initial assessment and six-year follow-up of Methods, it is stated that typical LSS symptoms were judged by LSS-SSHQ, so I inserted the following sentence in the result. Is this acceptable for you?
Two-hundred and thirty-two follow-up subjects were divided into two groups, typical LSS symptoms group and no typical symptoms group at the initial survey, and changes in typical LSS symptoms at follow-up were examined. In the initial typical LSS-positive group, 13 of 52 subjects (25%) remained typical LSS-positive at the six-year follow-up, whereas 39 of 52 (75%) were reclassified as typical LSS-negative.・・・
Discussion:
Page 6, Ln 181-187: The reasons provided are unclear. Please provide more detail.
Reply: Thank you very much for comments. We tried to explain in more detail as following. Are these acceptable for you?
In the present study, severe dural tube compression such as less than 25mm2 and 25-49.9mm2 of DCSA did not influence on both persist and developed typical LSS symptom for six years. For example, in the < 25mm2 of DCSA group, persistent typical LSS symptoms were present in 12.5% (2/16) and newly developed typical LSS symptoms occurred in 30% (3/10) (Table 3).Similarly, the number of intervertebral discs whose DCSA was less than 50mm2 did not show any tendency of maintaining and developing typical LSS symptoms. For example, in more than three intervertebral disc level less than 50mm2 of DCSA group, persistent typical LSS symptoms were present in 0% (0/5) and newly developed typical LSS symptoms occurred in20% (1/5) (Table 4). Moreover, according to a multivariate logistic regression analysis of four models, magnitude of dural tube compression was not a predictive factor for the presence of typical LSS symptoms at the six-year follow-up (Table 5).

Reviewer 2 Report
The idea was really good. In the last part of the paper you analyzed in a really clear way its limitations. I'd like to find some objective data, such as electromyiographs.
Author Response
Thank you very much for kind and useful comments to improve our manuscript. We considered the reviewer’s intentions and made corrections to improve the paper.
Reviewer 2.
The idea was really good. In the last part of the paper you analyzed in a really clear way its limitations. I'd like to find some objective data, such as electromyiographs.
Reply: Thank you very much for comments. Your comment encourages our research team to continue this type of research. Unfortunately, there was no objective data such as electromyographs. According to your comments, we add one sentence in Limitation section.
Third, LSS-SSHQ could not evaluate the severity of LSS symptoms, just define the presence or absence of typical LSS symptoms. In addition, there was no objective data such as those obtained by electromyography

Reviewer 3 Report
In my opinion , the narrowing of the spinal canal is not a full determinant of all symptoms of the lumbar spinal stenosis (LSS). The purpose of the study was to reveal whether magnitude of dural tube compression influence on the development or presence LSS symptoms at the six-year follow-up or not.
This prospective cohort study reported data to demonstrate that the magnitude of dural tube compression do not influence the presence of typical LSS symptoms. In my opinion the study is informative for JCM readers and deserves attention. I suggest English revision for typos and some sentences not completely clear.
Author Response
Reviewer 3.
In my opinion, the narrowing of the spinal canal is not a full determinant of all symptoms of the lumbar spinal stenosis (LSS). The purpose of the study was to reveal whether magnitude of dural tube compression influence on the development or presence LSS symptoms at the six-year follow-up or not.
This prospective cohort study reported data to demonstrate that the magnitude of dural tube compression do not influence the presence of typical LSS symptoms. In my opinion the study is informative for JCM readers and deserves attention. I suggest English revision for typos and some sentences not completely clear.
Reply: Thank you very much for your comments. All of our research member thank reviewer 3 because reviewer 3 understands the value of our research. We tried to improve the English text following the advice of a professional English translator. If it is approved for publication in JCM, we would like to follow the advice of the editor of JCM and make further improvements.
Reviewer 4 Report
This study revealed magnitude of dural tube compression influence on the development or presence LSS symptoms at the six-year follow-up. They analyzed the relationship between magnitude of dural tube compression evaluated by dural tube cross-sectional area (DCSA) in the initial assessment and the time course of typical LSS symptoms for six-year duration.
- The conclusion that “severe dural tube compression could not detect typical LSS-symptom development and occurrence of surgery during six-year period” is not new. Although the authors haven done thorough work, by compared with the authors previous study “Magnitude of dural tube compression does not show a predictive value for symptomatic lumbar spinal stenosis for 1-year follow-up: a prospective cohort study in the community”, they have not drawn any novel conclusion but only increase the follow-up period, and the conclusion would not provoke further discussion or investigation on the part of the reader.
- The authors need to discuss the reasons about “the score below the normative RDQ score and LSS at the initial analysis had statistically significant influence on the presence of LSS symptoms at the one-year follow-up but not at the six-year follow-up” based on the multivariate logistic regression analysis.
- The authors mentioned six-year follow-up rate is 50.5%, but how many patients were excluded during the follow-up? How many patients were lost follow-up?
- Line 38, 44, 106 and 168 cited the incorrect references.
Author Response
Thank you very much for kind and useful comments to improve our manuscript. We considered the reviewer’s intentions and made corrections to improve the paper.
Reviewer 4.
This study revealed magnitude of dural tube compression influence on the development or presence LSS symptoms at the six-year follow-up. They analyzed the relationship between magnitude of dural tube compression evaluated by dural tube cross-sectional area (DCSA) in the initial assessment and the time course of typical LSS symptoms for six-year duration.
1. The conclusion that “severe dural tube compression could not detect typical LSS-symptom development and occurrence of surgery during six-year period” is not new. Although the authors haven done thorough work, by compared with the authors previous study “Magnitude of dural tube compression does not show a predictive value for symptomatic lumbar spinal stenosis for 1-year follow-up: a prospective cohort study in the community”, they have not drawn any novel conclusion but only increase the follow-up period, and the conclusion would not provoke further discussion or investigation on the part of the reader.
Reply: Thank you very much for your critical comments. As you pointed out, the results of this study, in which the follow-up period was extended to 6 years, were the same as the 1-year follow-up period. However, our research team believes that it is very important that the results of the 6-year follow-up are the same as the 1-year follow-up. The reason is that even with the advanced anatomical findings of spinal stenosis, the 6-year period does not always result in symptoms. This means that asymptomatic spinal stenosis does not need to be actively operated, which can lead to a reduction in unnecessary surgery.
2. The authors need to discuss the reasons about “the score below the normative RDQ score and LSS at the initial analysis had statistically significant influence on the presence of LSS symptoms at the one-year follow-up but not at the six-year follow-up” based on the multivariate logistic regression analysis.
Reply: Thank you very much for your important comments. We try to explain our speculation. According to the results of a 6-year follow-up of typical LSS symptoms that did not include imaging findings, the factors that influence the presence of typical LSS symptoms after 6 years were the presence of typical LSS symptoms and low norm-based RDQ score at the initial assessment (Ref. Medicina 2021, 57, 1116). On the other hand, in this study including MRI findings, the presence of typical LSS symptoms, norm-based RDQ score and the degree of dural tube compression in the initial assessment did not affect the presence of typical LSS symptoms after 6 years. The difference between the results of these two studies mean that the degree of dural tube compression is strong enough to cancel out the positive effects of the presence of typical LSS symptoms and low norm-based RDQ score at the initial assessment to predict the presence of LSS symptoms after 6years. Therefore, dural tube compression strongly suggests that it does not predict the presence of typical LSS symptoms after 6 years. So, we add the sentences in Discussion; “This was the consistent with the one-year follow-up [5]. In contrast, the presence of typical LSS symptoms and low RDQ scores at the initial survey were predictors of the presence of typical LSS symptoms at the one-year follow-up, but not at the six-year follow-up. This means that the degree of dural tube compression is strong enough to cancel out the positive effects of both the presence of typical LSS symptoms and low norm-based RDQ score at the initial assessment predicting the presence of LSS symptoms after six years. Therefore, dural tube compression strongly is not a predictor of the presence of typical LSS symptoms after six years.
3. The authors mentioned six-year follow-up rate is 50.5%, but how many patients were excluded during the follow-up? How many patients were lost follow-up?
Reply: Thank you very much for your comment. A questionnaire was mailed to 459 people who participated in the initial survey, and 232 people who responded to LSS-SSHQ and 6 years of lumbar surgery history were targeted. Therefore, the follow-up rate was 232/459 = 50.5%. So, we changed the sentence in Methods; In this follow-up survey, 232 subjects (male: 73, female: 159) of 459 subjects who participated in the initial survey responded and were assessed for typical LSS symptoms using LSS-SSHQ and six years of lumbar surgery history (follow-up rate 50.5%) (Table 1).
4. Line 38, 44, 106 and 168 cited the incorrect references.
Reply: Thank you very much for your comment. We apologize for the incorrect article number quoted. This is because the first paper cited in the automatic number choreography was 2 instead of 1. This is a completely careless mistake.
